# Ultrametric Fitting by Gradient Descent

**Giovanni Chierchia**[*]
Université Paris-Est, LIGM (UMR 8049)
CNRS, ENPC, ESIEE Paris, UPEM
F-93162, Noisy-le-Grand, France
giovanni.chierchia@esiee.fr

**Benjamin Perret**[*]
Université Paris-Est, LIGM (UMR 8049)
CNRS, ENPC, ESIEE Paris, UPEM
F-93162, Noisy-le-Grand, France
benjamin.perret@esiee.fr

## Abstract

We study the problem of fitting an ultrametric distance to a dissimilarity graph in the context of hierarchical cluster analysis. Standard hierarchical clustering methods are specified procedurally, rather than in terms of the cost function to be optimized. We aim to overcome this limitation by presenting a general optimization framework for ultrametric fitting. Our approach consists of modeling the latter as a constrained optimization problem over the continuous space of ultrametrics. So doing, we can leverage the simple, yet effective, idea of replacing the ultrametric constraint with a min-max operation injected directly into the cost function. The proposed reformulation leads to an unconstrained optimization problem that can be efficiently solved by gradient descent methods. The flexibility of our framework allows us to investigate several cost functions, following the classic paradigm of combining a data fidelity term with a regularization. While we provide no theoretical guarantee to find the global optimum, the numerical results obtained over a number of synthetic and real datasets demonstrate the good performance of our approach with respect to state-of-the-art agglomerative algorithms. This makes us believe that the proposed framework sheds new light on the way to design a new generation of hierarchical clustering methods. Our code is made publicly available at https://github.com/PerretB/ultrametric-fitting.

## 1 Introduction

Ultrametrics provide a natural way to describe a recursive partitioning of data into increasingly finer clusters, also known as hierarchical clustering [1]. Ultrametrics are intuitively represented by dendrograms, *i.e.*, rooted trees whose leaves correspond to data points, and whose internal nodes represent the clusters of its descendant leaves. In topology, this corresponds to a metric space in which the usual triangle inequality is strengthened by the ultrametric inequality, so that every triple of points forms an isosceles triangle, with the two equal sides at least as long as the third side. The main question investigated in this article is: « *How well can we construct an ultrametric to fit the given dissimilarity data?* » This is what we refer to as ultrametric fitting.

Ultrametric fitting can be traced back to the early work on numerical taxonomy [2] in the context of phylogenetics [3]. Several well-known algorithms originated in this field, such as single linkage [4], average linkage [5], and Ward method [6]. Nowadays, there exists a large literature on ultrametric fitting, which can be roughly divided in four categories: agglomerative and divisive greedy heuristics [7–13], integer linear programming [14–16], continuous relaxations [17–20], and probabilistic formulations [21–23]. Our work belongs to the family of continuous relaxations.

The most popular methods for ultrametric fitting probably belong to the family of agglomerative heuristics. They follow a bottom-up approach, in which the given dissimilarity data are sequentially

---

[*]Both authors contributed equally.

merged through some specific strategy. But since the latter is specified procedurally, it is usually hard to understand the objective function being optimized. In this regard, several recent works [9, 16, 19, 11, 24] underlined the importance to cast ultrametric fitting as an optimization problem with a well-defined cost function, so as to better understand how the ultrametric is built.

Recently, Dasgupta [9] introduced a cost function for evaluating an ultrametric, and proposed an heuristic to approximate its optimal solution. The factor of this approximation was later improved by several works, based on a linear programming relaxation [16], a semidefinite programming relaxation [19], or a recursive $\phi$-sparsest cut algorithm [24]. Along similar lines, it was shown that average linkage provides a good approximation of the optimal solution to Dasgupta cost function [25, 26]. Closer to our approach, a differentiable relaxation inspired by Dasgupta cost function was also proposed [20]. Moreover, a regularization for Dasgupta cost function was formulated in the context of semi-supervised clustering [23, 27], based on triplet constraints provided by the user.

More generally, the problem of finding the closest ultrametric to dissimilarity data was extensively studied through linear programming relaxations [18] and integer linear programming [15]. A special case of interest arises when the dissimilarities are specified by a planar graph, which is a natural occurrence in image segmentation [28–31]. By exploiting the planarity of the input graph, a tight linear programming relaxation can be derived from the minimum-weight multi-cut problem [14]. There exist many other continuous relaxations of discrete problems in the specific context of image segmentation [32–37], but they typically aim at a flat representation of data, rather than hierarchical.

**Contribution.** We propose a general optimization framework for ultrametric fitting based on gradient descent. Our approach consists of optimizing a cost function over the continuous space of ultrametrics, where the ultrametricity constraint is implicitly enforced by a min-max operation. We demonstrate the versatility of our approach by investigating several cost functions:

1. the *closest-ultrametric* fidelity term, which expresses that the fitted ultrametric should be close to the given dissimilarity graph;
2. the *cluster-size* regularization, which penalizes the presence of small clusters in the upper levels of the associated hierarchical clustering;
3. the *triplet* regularization for semi-supervised learning, which aims to minimize the intra-class distance and maximize the inter-class distance;
4. the *Dasgupta* fidelity term, which is a continuous relaxation of Dasgupta cost function expressing that the fitted ultrametric should associate large dissimilarities to large clusters.

We devise efficient algorithms with automatic differentiation in mind, and we show that they scale up to millions of vertices on sparse graphs. Finally, we evaluate the proposed cost functions on synthetic and real datasets, and we show that they perform as good as Ward method and semi-supervised SVM.

## 2 Ultrametric fitting

Central to this work is the notion of *ultrametric*, a special kind of metric that is equivalent to hierarchical clustering [38]. Formally, an ultrametric $d\colon V \times V \to \mathbb{R}_+$ is a metric on a space $V$ in which the triangle inequality is strengthen by the *ultrametric inequality*, defined as

$$(\forall (x, y, z) \in V^3) \qquad d(x, y) \leq \max\{d(x, z), d(z, y)\}. \tag{1}$$

The notion of ultrametric can also be defined on a connected (non-complete) graph $\mathcal{G} = (V, E)$ with non-negative edge weights $w \in \mathcal{W}$, where $\mathcal{W}$ denotes the space of functions from $E$ to $\mathbb{R}_+$. In this case, the distance is only available between the pairs of vertices in $E$, and the ultrametric constraint must be defined over the set of cycles $\mathcal{C}$ of $\mathcal{G}$ as follows:

$$u \in \mathcal{W} \text{ is an ultrametric}^2 \text{ on } \mathcal{G} \qquad \Leftrightarrow \qquad (\forall C \in \mathcal{C}, \forall e \in C) \quad u(e) \leq \max_{e' \in C \setminus \{e\}} u(e'). \tag{2}$$

Note that an ultrametric $u$ on $\mathcal{G}$ can be extended to all the pairs of vertices in $V$ through the min-max distance on $u$, which is defined as

$$(\forall (x, y) \in V^2) \qquad d_u(x, y) = \min_{P \in \mathcal{P}_{xy}} \max_{e' \in P} u(e'), \tag{3}$$

where $\mathcal{P}_{xy}$ denotes the set of all paths between the vertices $x$ and $y$ of $\mathcal{G}$. This observation allows us to compactly represent ultrametrics as weight functions $u \in \mathcal{W}$ on sparse graphs $\mathcal{G}$, instead of more costly pairwise distances. Figure 1 shows an example of ultrametric and its possible representations.

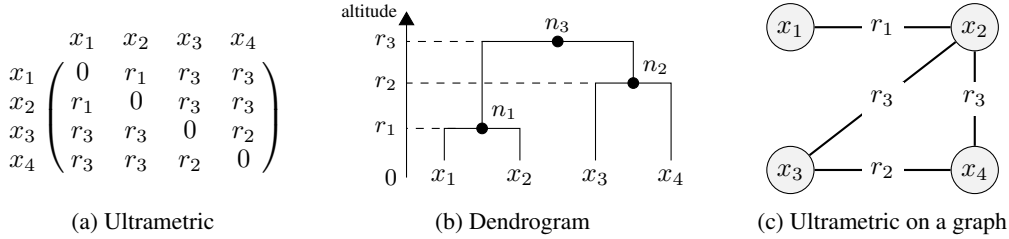

|       | $x_1$ | $x_2$ | $x_3$ | $x_4$ |
|-------|-------|-------|-------|-------|
| $x_1$ | 0     | $r_1$ | $r_3$ | $r_3$ |
| $x_2$ | $r_1$ | 0     | $r_3$ | $r_3$ |
| $x_3$ | $r_3$ | $r_3$ | 0     | $r_2$ |
| $x_4$ | $r_3$ | $r_3$ | $r_2$ | 0     |

|         |         |         |
|---------|---------|---------|
| (a) Ultrametric | (b) Dendrogram | (c) Ultrametric on a graph |

Figure 1: Ultrametric $d$ on $\{x_1, x_2, x_3, x_4\}$ given by the dissimilarity matrix (a), and represented by the dendrogram (b) and the graph (c). Two elements $x_i$ and $x_j$ merge at the altitude $r_k = d(x_i, x_j)$ in the dendrogram, and the corresponding node is the lowest common ancestor (l.c.a.) of $x_i$ and $x_j$. For example, the l.c.a. of $x_1$ and $x_2$ is the node $n_1$ at altitude $r_1$, hence $d(x_1, x_2) = r_1$; the l.c.a. of $x_1$ and $x_3$ is the node $n_3$ at altitude $r_3$, hence $d(x_1, x_3) = r_3$. The graph (c) with edge weights $u$ leads to the ultrametric (a) via the min-max distance $d_u$ defined in (3). For example, all the paths from $x_1$ to $x_3$ contain an edge of weight $r_3$ which is maximal, and thus $d_u(x_1, x_3) = r_3$.

**Notation.** The dendrogram associated to an ultrametric $u$ on $\mathcal{G}$ is denoted by $T_u$ [38]. It is a rooted tree whose leaves are the elements of $V$. Each tree node $n \in T_u$ is the set composed by all the leaves of the sub-tree rooted in $n$. The altitude of a node $n$, denoted by $\text{alt}_u(n)$, is the maximal distance between any two elements of $n$: *i.e.*, $\text{alt}_u(n) = \max \{u(e_{xy}) \mid x, y \in n \text{ and } e_{xy} \in E\}$. The size of a node $n$, denoted by $|n|$, is the number of leaves contained in $n$. For any two leaves $x$ and $y$, the lowest common ancestor of $x$ and $y$, denoted $\text{lca}_u(x, y)$, is the smallest node of $T_u$ containing both $x$ and $y$.

## 2.1 Optimization framework

Our goal is to find the ultrametric that "best" represents the given edge-weighted graph. We propose to formulate this task as a constrained optimization problem involving an appropriate cost function $J: \mathcal{W} \to \mathbb{R}$ defined on the (continuous) space of distances $\mathcal{W}$, leading to

$$\underset{u \in \mathcal{W}}{\text{minimize}} \quad J(u; w) \quad \text{s.t.} \quad u \text{ is an ultrametric on } \mathcal{G}. \tag{4}$$

The ultrametricity constraint is highly nonconvex and cannot be efficiently tackled with standard optimization algorithms. We circumvent this issue by replacing the constraint with an operation injected directly into the cost function. The idea is that the ultrametricity constraint can be enforced implicitly through the operation that computes the subdominant ultrametric, defined as the largest ultrametric below the given dissimilarity function. One way to compute the subdominant ultrametric is through the min-max operator $\Phi_{\mathcal{G}}: \mathcal{W} \to \mathcal{W}$ defined by

$$(\forall \tilde{w} \in \mathcal{W}, \forall e_{xy} \in E) \qquad \Phi_{\mathcal{G}}(\tilde{w})(e_{xy}) = \min_{P \in \mathcal{P}_{xy}} \max_{e' \in P} \tilde{w}(e'), \tag{5}$$

where $\mathcal{P}_{xy}$ is defined as in (3). Then, Problem (4) can be rewritten as

$$\underset{\tilde{w} \in \mathcal{W}}{\text{minimize}} \ J\big(\Phi_{\mathcal{G}}(\tilde{w}); w\big). \tag{6}$$

Since the min-max operator is sub-differentiable (see (15) in Section 3), the above problem can be optimized by gradient descent, as long as $J$ is sub-differentiable. This allows us to devise Algorithm 1.

Note that the mix-max operator already proved useful in image segmentation to define a structured loss function for end-to-end supervised learning [28, 31]. The goal was however to find a flat segmentation rather than a hierarchical one. To the best of our knowledge, we are the first to use the mix-max operator within an optimization framework for ultrametric fitting.

## 2.2 Closest ultrametric

A natural goal for ultrametric fitting is to find the closest ultrametric to the given dissimilarity graph. This task fits nicely into Problem (4) by setting the cost function to the sum of squared errors between the sought ultrametric and the edge weights of the given graph, namely

$$J_{\text{closest}}(u; w) = \sum_{e \in E} \big(u(e) - w(e)\big)^2. \tag{7}$$

---

**Algorithm 1** Solution to the ultrametric fitting problem defined in (4).

---

**Require:** Graph $\mathcal{G} = (V, E)$ with edge weights $w$
1: $\tilde{w}^{[0]} \leftarrow w$
2: **for** $t = 0, 1, \ldots$ **do**
3:      $g^{[t]} \leftarrow$ gradient of $J\big(\Phi_{\mathcal{G}}(\cdot);\, w\big)$ evaluated at $\tilde{w}^{[t]}$
4:      $\tilde{w}^{[t+1]} \leftarrow$ update of $\tilde{w}^{[t]}$ using $g^{[t]}$
5: **return** $\Phi_{\mathcal{G}}(\tilde{w}^{[\infty]})$

---

Although the exact minimization of this cost function is a NP-hard problem [40], the proposed optimization framework allows us to compute an approximate solution. Figure 2 shows the ultrametric computed by Algorithm 1 with $J_{\text{closest}}$ for an illustrative example of hierarchical clustering.

A common issue with the closest ultrametric is that small clusters might branch very high in the dendrogram. This is also true for average linkage and other agglomerative methods. Such kind of ultrametrics are undesirable, because they lead to partitions containing very small clusters at large scales, as clearly shown in Figures 2b-2c. We now present two approaches to tackle this issue.

## 2.3 Cluster-size regularization

To fight against the presence of small clusters at large scales, we need to introduce a mechanism that pushes down the altitude of nodes where such incorrect merging occurs. This can be easily translated in our framework, as the altitude of a node corresponds to the ultrametric distance between its children. Specifically, we penalize the ultrametric distance proportionally to some non-negative coefficients that depend on the corresponding nodes in the dendrogram, yielding

$$J_{\text{size}}(u) = \sum_{e_{xy} \in E} \frac{u(e_{xy})}{\gamma_u(\text{lca}_u(x, y))}. \tag{8}$$

Here above, the $\gamma$ coefficients play an essential role: they must be small for the nodes that need to be pushed down, and large otherwise. We thus rank the nodes by the size of their smallest child, that is

$$(\forall n \in T_u) \qquad \gamma_u(n) = \min\left\{|c|, c \in \text{Children}_u(n)\right\}, \tag{9}$$

where $\text{Children}_u(n)$ denotes the children of a node $n$ in the dendrogram $T_u$ associated to the ultrametric $u$ on $\mathcal{G}$. Figure 2d shows the ultrametric computed by Algorithm 1 with $J_{\text{closest}} + J_{\text{size}}$. The positive effect of this regularization can be appreciated by observing that small clusters are no longer branched very high in the dendrogram.

## 2.4 Triplet regularization

Triplet constraints [23, 27] provide an alternative way to penalize small clusters at large scales. Like in semi-supervised classification, we assume that the labels $\mathcal{L}_v$ of some data points $v \in V$ are known, and we build a set of triplets according to the classes they belong to:

$$\mathcal{T} = \left\{(\text{ref}, \text{pos}, \text{neg}) \in V^3 \mid \mathcal{L}_{\text{ref}} = \mathcal{L}_{\text{pos}} \quad \text{and} \quad \mathcal{L}_{\text{ref}} \neq \mathcal{L}_{\text{neg}}\right\}. \tag{10}$$

These triplets provide valuable information on how to build the ultrametric. Intuitively, we need a mechanism that reduces the ultrametric distance within the classes, while increasing the ultrametric distance between different classes. This can be readily expressed in our framework with a regularization acting on the altitude of nodes containing the triplets, leading to

$$J_{\text{triplet}}(u) = \sum_{(\text{ref}, \text{pos}, \text{neg}) \in \mathcal{T}} \max\{0, \alpha + d_u(\text{ref}, \text{pos}) - d_u(\text{ref}, \text{neg})\}. \tag{11}$$

Here above, the constant $\alpha > 0$ represents the minimum prescribed distance between different classes. Figure 2e shows the ultrametric computed by Algorithm 1 with $J_{\text{closest}} + J_{\text{triplet}}$.

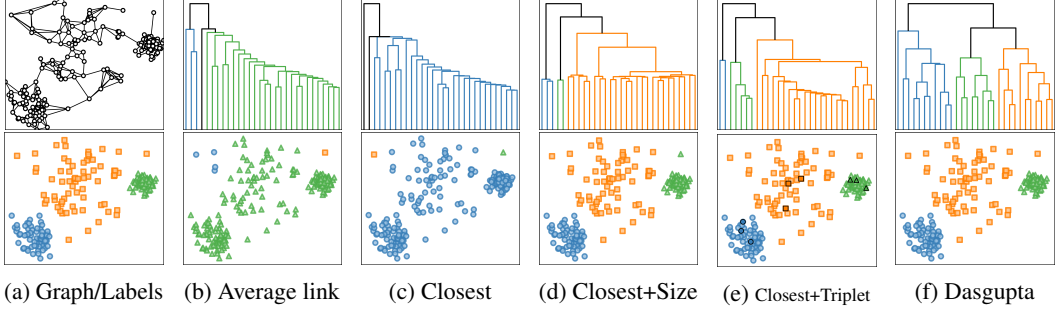

| (a) Graph/Labels | (b) Average link | (c) Closest | (d) Closest+Size | (e) Closest+Triplet | (f) Dasgupta |

Figure 2: Illustrative examples of hierarchical clustering. *Top row*: Ultrametrics fitted to the input graph; only the top-30 non-leaf nodes are shown in the dendrograms (all the others are contracted into leaves). *Bottom row*: Assignments obtained by thresholding the ultrametrics at three clusters. The detrimental effect of having "small clusters at large scales" can be observed in (b) and (c).

### 2.5 Dasgupta cost function

Dasgupta cost function [9] has recently gained traction in the seek of a theoretically grounded framework for hierarchical clustering [16, 11, 41, 24, 27]. However its minimization is known to be a NP-hard problem [9]. The intuition behind this function is that large clusters should be associated to large dissimilarities. The idea it then to minimize, for each edge $e \in E$, the size of the dendrogram node associated to $e$ divided by the weight of $e$, yielding

$$J_{\text{Dasgupta}}(u; w) = \sum_{e_{xy} \in E} \frac{|\text{lca}_u(x, y)|}{w(e_{x,y})}. \tag{12}$$

However, we cannot directly use (12) in our optimization framework, as the derivative of $|\text{lca}_u(x, y)|$ with respect to the underlying ultrametric $u$ is equal to 0 almost everywhere. To solve this issue, we propose a *soft-cardinal* measure of a node that is differentiable w.r.t. the associated ultrametric $u$. Let $n$ be a node of the dendrogram $T_u$, and let $\{x\} \subseteq n$ be a leaf of the sub-tree rooted in $n$. We observe that the cardinal of $n$ is equal to the number of vertices $y \in V$ such that the ultrametric distance $d_u(x, y)$ between $x$ and $y$ is strictly lower than the altitude of $n$, namely

$$|n| = \sum_{y \in V} H(\text{alt}_u(n) - d_u(x, y)), \tag{13}$$

where $H$ is the Heaviside function. By replacing $H$ with a continuous approximation, such as a sigmoid function, we provide a soft-cardinal measure of a node of $T_u$ that is differentiable with respect to the ultrametric $u$. Figure 2f shows the ultrametric computed by Algorithm 1 with $J_{\text{Dasgupta}}$.

Note that a differentiable cost function inspired by Dasgupta cost function was proposed in [20]. This function replaces the node size by a parametric probability measure which is optimized over a fixed tree. This is fundamentally different from our approach, where the proposed measure is a continuous relaxation of the node size, and it is directly optimized over the ultrametric distance.

## 3 Algorithms

In this section, we present a general approach to efficiently compute the various terms appearing in the cost functions introduced earlier. All the proposed algorithms rely on some properties of the *single linkage (agglomerative) clustering*, which is a dual representation of the subdominant ultrametric. We perform a detailed analysis of the algorithm used to compute the subdominant ultrametric. The other algorithms can be found in the supplemental material.

Single-linkage clustering can be computed similarly to a minimum spanning tree (m.s.t.) with Kruskal's algorithm, by sequentially processing the edges of the graph in non decreasing order, and merging the clusters located at the extremities of the edge, when a m.s.t. edge is found. One consequence of this approach is that each node $n$ of the dendrogram representing the single linkage clustering is canonically associated to an edge of the m.s.t. (see Figure 3a), which is denoted by $\sigma(n)$.

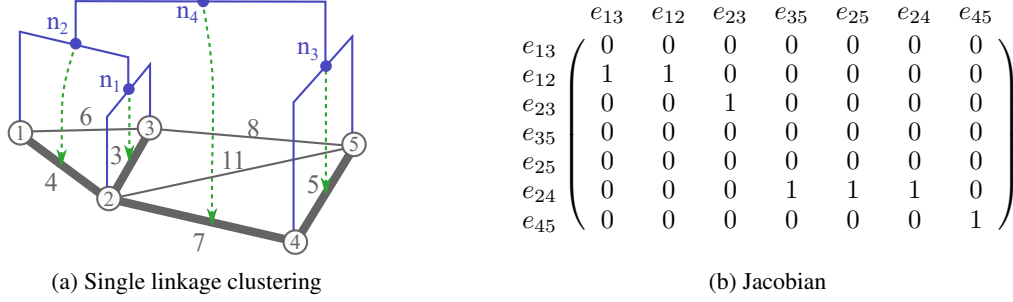

(a) Single linkage clustering         (b) Jacobian

Figure 3: Each node of the single linkage clustering (in blue) of the graph (in grey) is canonically associated (green dashed arrows) to an edge of a minimum spanning tree of the graph (thick edges): this edge is the pass edge between the leaves of the two children of the node. Edges are numbered from 1 to $M$ (number of edges). The $i$-th column of the Jacobian matrix of the $\Phi$ operator is equal to the indicator vector denoting the pass edge holding the maximal value of the min-max path between the two extremities of the $i$-th edge. The pass edge can be found efficiently in the single linkage clustering using the l.c.a. operation and the canonical link between the nodes and the m.s.t. edges. For example, the l.c.a. of the vertices 3 and 5 linked by the 4-th edge $e_{35}$ is the node $n_4$, which is canonically associated to the 6-th edge $e_{24}$ ($\sigma(n_4) = e_{24}$): we thus have $J_{6,4} = 1$.

In the following, we assume that we are working with a sparse graph $\mathcal{G} = (V, E)$, where $\mathcal{O}(|E|) = |V|$. The number of vertices (resp. edges) of $\mathcal{G}$ is denoted by $N$ (resp. $M$). For the ease of writing, we denote the edge weights as vectors of $\mathbb{R}^M$. The dendrogram corresponding to the single-linkage clustering of the graph $\mathcal{G}$ with edge weights $\tilde{w} \in \mathcal{W}$ is denoted by $\mathrm{SL}(\tilde{w})$.

### 3.1 Subdominant ultrametric

To obtain an efficient and automatically differentiable algorithm for computing the subdominant ultrametric, we observe that the min-max distance between any two vertices $x, y \in V$ is given by the weight of the *pass edge* between $x$ and $y$. This is the edge holding the maximal value of the min-max path from $x$ to $y$, and an arbitrary choice is made if several pass edges exist. Moreover, the pass edge between $x$ and $y$ corresponds to the l.c.a. of $x$ and $y$ in the single linkage clustering of $(\mathcal{G}, \tilde{w})$ (see Figure 3a). Equation (5) can be thus rewritten as

$$(\forall \tilde{w} \in \mathcal{W}, \forall e_{xy} \in E) \qquad \Phi_{\mathcal{G}}(\tilde{w})(e_{xy}) = \tilde{w}(e_{xy}^{\mathrm{mst}}) \quad \text{with} \quad e_{xy}^{\mathrm{mst}} = \sigma(\mathrm{lca}_{\mathrm{SL}(\tilde{w})}(x, y)). \qquad (14)$$

The single-linkage clustering can be computed in time $\mathcal{O}(N \log N)$ with a variant of Kruskal's minimum spanning tree algorithm [4, 42]. Then, a fast algorithm allows us to compute the l.c.a. of two nodes in constant time $\mathcal{O}(1)$, thanks to a linear time $\mathcal{O}(N)$ preprocessing of the tree [43]. The subdominant ultrametric can thus be computed in time $\mathcal{O}(N \log N)$ with Algorithm 2. Moreover, the dendrogram associated to the ultrametric returned by Algorithm 2 is the tree computed on line 2.

Note that Algorithm 2 can be interpreted as a special max pooling applied to the input tensor $w$, and can be thus automatically differentiated. Indeed, a sub-gradient of the min-max operator $\Phi$ at a given edge $e_{xy}$ is equal to 1 on the pass edge between $x$ and $y$ and 0 elsewhere. Then, the Jacobian of the min-max operator $\Phi$ can be written as the matrix composed of the indicator column vectors giving the position of the pass edge associated to the extremities of each edge in $E$:

$$\frac{\partial \Phi(\tilde{w})}{\partial \tilde{w}} = \left[ \mathbb{1}_{\Phi_{\mathcal{G}}^*(\tilde{w}_1)}, \ldots, \mathbb{1}_{\Phi_{\mathcal{G}}^*(\tilde{w}_M)} \right], \qquad (15)$$

where $\Phi_{\mathcal{G}}^*(\tilde{w}_i)$ denotes the index of the pass edge between the two extremities of the $i$-th edge, and $\mathbb{1}_j$ is the column vector of $\mathbb{R}^M$ equals to 1 in position $j$, and 0 elsewhere (see Figure 3b).

### 3.2 Regularization terms

The cluster-size regularization defined in (8) can be implemented through the same strategy used in Algorithm 2, based on the single-linkage clustering and the fast l.c.a. algorithm, leading to a time complexity in $\mathcal{O}(N \log N)$. See supplemental material.

**Algorithm 2** Subdominant ultrametric operator defined in (5) with (14).

---
**Require:** Graph $\mathcal{G} = (V, E)$ with edge weights $w$
1: $u(e_{xy}) \leftarrow 0$ **for each** $e_{xy} \in E$                                             $\triangleright \mathcal{O}(N)$
2: $tree \leftarrow$ single-linkage$(\mathcal{G}, w)$                               $\triangleright \mathcal{O}(N \log N)$ with[4, 42]
3: preprocess l.c.a. on $tree$                                   $\triangleright \mathcal{O}(N)$ with [43]
4: **for each** edge $e_{xy} \in E$ **do**                                  $\triangleright \mathcal{O}(N)$
5:     $pass\_node \leftarrow \text{lca}_{tree}(x, y)$                        $\triangleright \mathcal{O}(1)$ with [43]
6:     $pass\_edge \leftarrow \sigma(pass\_node)$                   $\triangleright \mathcal{O}(1)$ see Figure 3a
7:     $u(e_{xy}) \leftarrow w(pass\_edge)$                        $\triangleright \mathcal{O}(1)$
8: **return** $u$

---

Furthermore, thanks to Equation (14), the triplet regularization defined in (11) can be written as

$$J_{\text{triplet}}(u) = \sum_{(\text{ref},\text{pos},\text{neg})\in\mathcal{T}} \max\{0, \alpha + u\big(\sigma(\text{lca}_u(\text{ref}, \text{pos}))\big) - u\big(\sigma(\text{lca}_u(\text{ref}, \text{neg}))\big)\}. \quad (16)$$

This can be implemented with a time complexity in $\mathcal{O}(|\mathcal{T}| + N \log N)$. See supplemental material.

### 3.3 Dasgupta cost function

The soft cardinal of a node $n$ of the tree $T_u$ as defined in (13) raises two issues: the arbitrary choice of the reference vertex $x$, and the quadratic time complexity $\Theta(N^2)$ of a naive implementation. One way to get rid of the arbitrary choice of $x$ is to use the two extremities of the edge $\sigma(n)$ canonically associated to $n$. To efficiently compute (13), we can notice that, if $c_1$ and $c_2$ are the children of $n$, then the pass edge between any element $x$ of $c_1$ and any element $y$ of $c_2$ is equal to edge $\sigma(n)$ associated to $n$. This allows us to define $\text{card}(n)$ as the relaxation of $|n|$ in (13), by replacing the Heaviside function $H$ with the sigmoid function $\ell$, leading to

$$\begin{aligned}
\text{card}(n) &= \frac{1}{2} \sum_{x\in\sigma(n)} \sum_{y\in V} \ell(\text{alt}_u(n) - d_u(x, y)) \\
&= \frac{1}{2} \sum_{x\in\sigma(n)} \Big( \ell(\text{alt}_u(n) - d_u(x, x)) + \sum_{y\in V\setminus\{x\}} \ell(\text{alt}_u(n) - d_u(x, y)) \Big) \\
&= \frac{1}{2} \sum_{x\in\sigma(n)} \Big( \ell(\text{alt}_u(n)) + \sum_{y\in\mathcal{A}(x)} \sum_{z\in c_{\hat{x}}(y)} \ell(\text{alt}_u(n) - d_u(x, z)) \Big) \\
&= \frac{1}{2} \sum_{x\in\sigma(n)} \Big( \ell(\text{alt}_u(n)) + \sum_{y\in\mathcal{A}(x)} \sum_{z\in c_{\hat{x}}(y)} \ell(\text{alt}_u(n) - \text{alt}_u(y)) \Big) \\
&= \frac{1}{2} \sum_{x\in\sigma(n)} \Big( \ell(\text{alt}_u(n)) + \sum_{y\in\mathcal{A}(x)} |c_{\hat{x}}(y)| \ell(\text{alt}_u(n) - \text{alt}_u(y)) \Big) \\
&= \frac{1}{2} \sum_{x\in\sigma(n)} \Big( \ell\big(u(\sigma(n))\big) + \sum_{y\in\mathcal{A}(x)} |c_{\hat{x}}(y)| \ell\big(u(\sigma(n)) - u(\sigma(y))\big) \Big), \quad (17)
\end{aligned}$$

where $\mathcal{A}(x)$ is the set of ancestors of $x$, and $c_{\hat{x}}(y)$ is the child of $y$ that does not contain $x$. The time complexity to evaluate (17) is dominated by the sum over the ancestors of $n$ which, in the worst case, is in the order of $\mathcal{O}(N)$, leading also to worst case time complexity of $\mathcal{O}(N^2)$. In practice, dendrograms are usually well balanced, and thus the number of ancestors of a node is in the order of $\mathcal{O}(\log N)$, yielding an empirical complexity in $\mathcal{O}(N \log N)$.

## 4 Experiments

In the following, we evaluate the proposed framework on two different setups. The first one aims to assess our continuous relaxation of the closest ultrametric problem with respect to the (almost) exact solution provided by linear programming on planar graphs [14]. The second one aims to compare the performance of our cost functions to the classical hierarchical and semi-supervised clustering

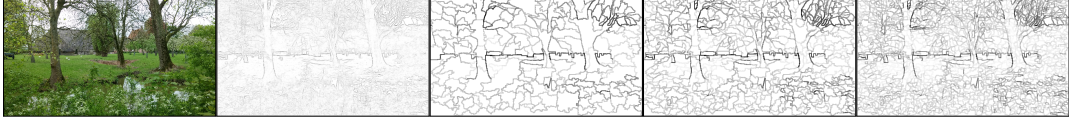

Figure 4: Test image, its gradient, and superpixel contour weights with 525, 1526, and 4434 edges.

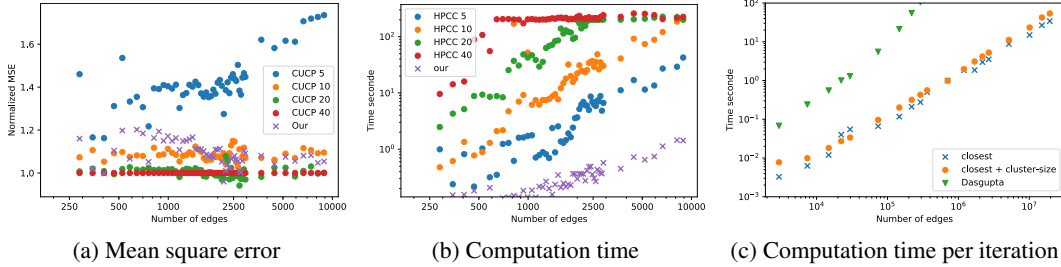

(a) Mean square error        (b) Computation time        (c) Computation time per iteration

Figure 5: Validation and computation time. Figureas (a) and (b): comparison between the CUCP algorithm [14] and the proposed gradient descent approach. For CUCP we tested different numbers of hierarchy levels (5, 10, 20 40) distributed evenly over the range of the input dissimilarity function. Figure (a) shows the final mean square error (normalized against CUCP 40) w.r.t. the number of edges in the tested graph. Figure (b) shows the run-time compared w.r.t. the number of edges in the tested graph (CUCP was capped at 200 seconds per instance). Figure (c) shows the computation time of the tested cost functions (one iteration of Algorithm 1) with respect to the number of edges in the graph.

methods. The implementation of our algorithms, based on Higra [44] and PyTorch [45] libraries, is available at `https://github.com/PerretB/ultrametric-fitting`.

**Framework validation** As Problem (6) is non-convex, there is no guarantee that the gradient descent method will find the global optimum. To assess the performance of the proposed framework, we use the algorithm proposed in [14], denoted by CUCP (Closest Ultrametric via Cutting Plane), as a baseline for the closest ultrametric problem defined in (7). Indeed, CUCP can provides an (almost) exact solution to the closest ultrametric problem for planar graphs based on a reformulation as a set of correlation clustering/multi-cuts problems with additional hierarchical constraints. However, CUCP requires to define a priori the set of levels which will compose the final hierarchy.

We generated a set of superpixels adjacency graphs of increasing scale from a high-resolution image (see Figure 4). The weight of the edge linking two superpixels is defined as the mean gradient value, obtained with [46], along the frontier between the two superpixels. The results presented in Figure 5 shows that the proposed approach is able to provide solutions close to the optimal ones (Figure 5a) using only a fraction of the time needed by the combinatorial algorithm (Figure 5b), and without any assumption on the input graph. The complete experimental setup is in the supplemental material.

The computation time of some combinations of cost terms are presented in Figure 5c. Note that, Algorithm 1 usually achieves convergence in about one hundred iterations (see supplemental material). *Closest* and *Closest+Size* can handle graphs with millions of edges. *Dasgupta* relaxation is computationally more demanding, which decreases the limit to a few hundred thousands of edges.

**Hierarchical clustering** We evaluate the proposed optimization framework on five datasets downloaded from the LIBSVM webpage,[3] whose size ranges from 270 to 1500 samples. For each dataset, we build a 5-nearest-neighbor graph, to which we add the edges of a minimum spanning tree to ensure the connectivity. Then, we perform hierarchical clustering on this graph, and we threshold the resulting ultrametric at the prescribed number of clusters. We divide our analysis in two sets of comparisons: hierarchical clustering (unsupervised), and semi-supervised clustering. To be consistent among the two types of comparisons, we use the classification accuracy as a measure of performance.

Figure 6a compares the performance of three hierarchical clustering methods. The baseline is "*Ward*" agglomerative method, applied to the pairwise distance matrix of each dataset. Average linkage

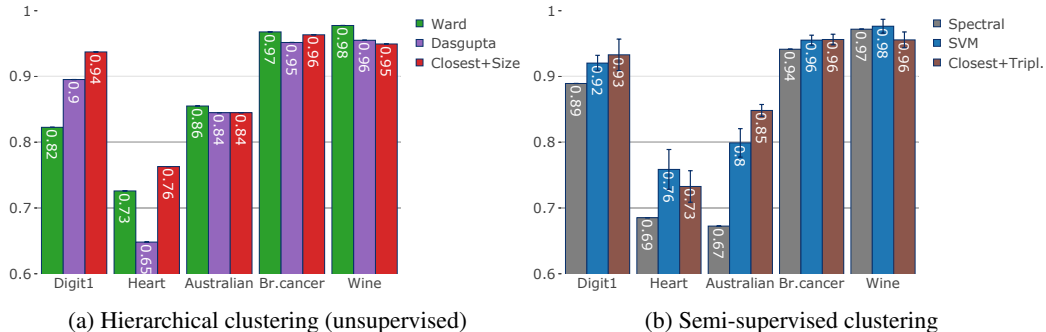

|                | (a) Hierarchical clustering (unsupervised) | (b) Semi-supervised clustering |
|----------------|--------------------------------------------|--------------------------------|

Figure 6: Performance on real datasets.

and closest ultrametric are not reported, as their performance is consistently worst. The "*Dasgupta*" method refers to Algorithm 1 with $J_{\text{Dasgupta}} + \lambda J_{\text{size}}$ and $\lambda = 1$. The "*Closest+Size*" method refers to Algorithm 1 with the cost function $J_{\text{closest}} + \lambda J_{\text{size}}$ and $\lambda = 10$. In both cases, the regularization is only applied to the top-10 dendogram nodes (see supplemental material). The results show that the proposed approach is competitive with Ward method (one of the best agglomerative heuristics). On the datasets Digit1 and Heart, "*Dasgupta*" performs slightly worse than "*Closest+Size*": this is partly due to the fact that our relaxation of the Dasgupta cost function is sensible to data scaling.

Figure 6b compares the performance of two semi-supervised clustering methods, and an additional unsupervised method. The first baseline is "*Spectral*" clustering applied to the Gaussian kernel matrix of each dataset. The second baseline is "*SVM*" classifier trained on the fraction of labeled samples, and tested on the remaining unlabeled samples. Between $10\%$ and $40\%$ of training samples are drawn from each dataset using a 10-fold scheme, and the cross-validated performance is reported in terms of mean and standard deviation. The "*Closest+Triplet*" method refers to Algorithm 1 with $J_{\text{closest}} + \lambda J_{\text{triplet}}$, $\lambda = 1$ and $\alpha = 10$. The results show that the triplet regularization performs comparably to semi-supervised SVM, which in turn performs better than spectral clustering.

## 5   Conclusion

We have presented a general optimization framework for fitting ultrametrics to sparse edge-weighted graphs in the context of hierarchical clustering. We have demonstrated that our framework can accommodate various cost functions, thanks to efficient algorithms that we have carefully designed with automatic differentiation in mind. Experiments carried on simulated and real data allowed us to show that the proposed approach provides good approximate solutions to well-studied problems.

The theoretical analysis of our optimization framework is beyond the scope of this paper. Nonetheless, we believe that statistical physics modelling [47] may be a promising direction for future work, based on the observation that ultrametricity is a physical property of spin-glasses [48–53]. Other possible extensions include the end-to-end learning of neural networks for hierarchical clustering, possibly in the context of image segmentation.

## 6   Acknowledgment

This work was partly supported by the INS2I JCJC project under grant 2019OSCI. We are deeply grateful to Julian Yarkony and Charless Fowlkes for sharing their code, and to Fred Hamprecht for many insightful discussions.

## Footnotes

[2] Some authors use different names, such as *ultrametric contour map* [29] or *saliency map* [39].

[3]`https://www.csie.ntu.edu.tw/~cjlin/libsvmtools/datasets/`

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
