[Supplementary Material]

# Ultrametric Fitting by Gradient Descent
## SUPPLEMENTAL

**Giovanni Chierchia**[*]
Laboratoire d'informatique Gaspard-Monge
ESIEE Paris, France
`giovanni.chierchia@esiee.fr`

**Benjamin Perret**[*]
Laboratoire d'informatique Gaspard-Monge
ESIEE Paris, France
`benjamin.perret@esiee.fr`

## 1   Average linkage approximates the closest ultrametric problem

To help understand how the closest ultrametric problem relates to average linkage, note that an ultrametric takes a finite set of non-negative values $\{r_1, \ldots, r_K\}$ with $K < |V|$. Hence, it can be represented as

$$(\forall e \in E) \qquad u(e) = \sum_{k=1}^{K} r_k \, x_k(e), \tag{1}$$

where $x_1, \ldots, x_K$ are functions from $E$ to $\{0, 1\}$ defining a hierarchical partition. In this setting, the cost function $J_{\text{closest}}$ boils down to

$$\widetilde{J}_{\text{closest}}(x, r; w) = \sum_{e \in E} \sum_{k=1}^{K} x_k(e) \big(r_k - w(e)\big)^2, \tag{2}$$

and, for a fixed hierarchical clustering $\bar{x}$, the optimal altitudes are given by

$$\bar{r}_k = \frac{\sum_{e \in E} \bar{x}_k(e) w(e)}{\sum_{e \in E} \bar{x}_k(e)}. \tag{3}$$

This is exactly the criterion used by average linkage to build a hierarchical clustering. We can thus argue that the latter provides an approximate solution to the closest ultrametric problem. As a matter of fact, average linkage and Algorithm 1 with $J_{\text{closest}}$ produce structurally similar ultrametrics, as shown in Figure 3 for illustrative examples of hierarchical clustering.

## 2   Algorithms

This section describes in detail the algorithms proposed to compute the cost terms associated to cluster-size regularization, triplet regularization, and Dapgusta cost function.

**Cluster-size regularization**   This regularization penalizes small clusters at large scales (see Figure 1a), and the associated cost is computed by Algorithm 1. It proceeds by first computing the size of the smallest child of each node of the tree. The size of each node can be trivially computed recursively from the leaves to the root in linear time. Then, it identifies the pass node associated to each edge thanks to he fast l.c.a. algorithm, and it deduces the individual cost for this edge. Note that we assume the weights $\gamma_u$ are constants, even though they depend on the variable $u$ being optimized. This allows us to simplify the gradient evaluation. Moreover, the algorithm presents an additional hyper-parameter $k$ for applying the regularization only to the top-$k$ dendrogram nodes. In the algorithm we denote by rank$(n)$, the rank of a node $n$ according to the ordering given by their altitudes (from highest to lowest values). The root of the tree is thus ranked 1, the second highest node is ranked 2 and so on.

---

[*]Both authors contributed equally.

(a) The cluster-size regularization $J_{\text{size}}$ pushes down the nodes of the hierarchical clustering having at least one small child. This corresponds to reducing the distance between the elements in the cluster, effectively preventing small clusters to appear at high altitudes.

(b) The triplet regularization $J_{\text{triplet}}$ pushes down the lowest common ancestor between elements of the same class (*i.e.*, it reduces the intra-class distance) and pushes up the lowest common ancestor between elements of different classes (*i.e.*, it increases the inter-class distance).

Figure 1: Intuitive interpretation of the proposed regularization schemes.

---

**Algorithm 1** Cluster-size regularization (7)

---

**Require:** Graph $\mathcal{G} = (V, E)$ with ultrametric $u$
**Require:** Parameter $k$ to apply regularization only on the top-$k$ nodes
**Require:** Output the cluster-size regularization value
1: $tree \leftarrow$ single-linkage$(\mathcal{G}, w)$          $\triangleright \mathcal{O}(N \log N)$ with[1, 2]
2: $area \leftarrow$ cardinal of each node of $tree$          $\triangleright \mathcal{O}(N)$
3: **for each** node $n$ of $tree$ from the leaves to the root (excluded) **do**          $\triangleright \mathcal{O}(N)$
4:      $min\_area\_children(n) \leftarrow \infty$          $\triangleright \mathcal{O}(1)$
5:      **for each** child $c$ of $n$ **do**          $\triangleright \mathcal{O}(1)$
6:          $min\_area\_children(n) \leftarrow \min(min\_area\_children(n), area(c))$          $\triangleright \mathcal{O}(1)$
7: preprocess l.c.a on $tree$          $\triangleright \mathcal{O}(N)$ with [3]
8: $reg \leftarrow 0$          $\triangleright \mathcal{O}(1)$
9: **for each** edge $e_{xy} \in E$ **do**          $\triangleright \mathcal{O}(N)$
10:      $pass\_node \leftarrow lca_{tree}(x, y)$          $\triangleright \mathcal{O}(1)$ with [3]
11:      **if** rank$(pass\_node) \leq k$ **then**          $\triangleright \mathcal{O}(1)$
12:          $reg \leftarrow reg + u(e_{xy})/min\_area\_children(pass\_node)$          $\triangleright \mathcal{O}(1)$
13: **return** $reg$

---

Note that in practice, the single linkage algorithm (as any agglomerative clustering method) naturally orders nodes according to this ordering and no extra computation is required. We set $k = 10$ in all our numerical experiments.

**Triplet regularization** This regularization for semi-supervised learning enforces triplet constraints (see Figure 1b), and the associated cost is computed by Algorithm 2, which is very similar to the subdominant ultrametric algorithm. For every triplet (ref, pos, neg), it searches for $n_1$ and $n_2$, the smallest clusters containing the pairs (ref, pos) and (ref, neg), with the fast l.c.a. algorithm. It then introduces a penalization if the altitude of $n_1$ (*i.e.*, the distance between ref and pos) is not small enough compared to the altitude of $n_2$ (*i.e.*, the distance between ref and neg).

**Dasgupta cost function** The difficulty in implementing the proposed relaxation of Dasgupta cost term lies in the soft-cardinal function defined in (16). The main function described in Algorithm 3 is similar to previously presented algorithm. The soft-cardinal function is computed by algorithm 4. As

---

**Algorithm 2** Triplet regularization (15)

---

**Require:** Graph $\mathcal{G} = (V, E)$ with ultrametric $u$
**Require:** Triplets $\mathcal{T} \subset \mathcal{V}^\ni$
**Require:** Margin $\alpha \in \mathbb{R}_+$
**Require:** Output the triplet regularization value
1: $tree \leftarrow$ single-linkage$(\mathcal{G}, w)$ &emsp;&emsp;&emsp;&emsp;&emsp; $\triangleright \mathcal{O}(N \log N)$ with[1, 2]
2: preprocess l.c.a on $tree$ &emsp;&emsp;&emsp;&emsp;&emsp;&emsp; $\triangleright \mathcal{O}(N)$ with [3]
3: $reg \leftarrow 0$ &emsp;&emsp;&emsp;&emsp;&emsp;&emsp;&emsp;&emsp;&emsp;&emsp; $\triangleright \mathcal{O}(1)$
4: **for each** $(\mathrm{ref}, \mathrm{pos}, \mathrm{neg}) \in \mathcal{T}$ **do** &emsp;&emsp;&emsp;&emsp;&emsp; $\triangleright \mathcal{O}(|\mathcal{T}|)$
5: &emsp; $pass\_node_1 \leftarrow lca_{tree}(ref, pos)$ &emsp;&emsp;&emsp;&emsp; $\triangleright \mathcal{O}(1)$ with [3]
6: &emsp; $pass\_node_2 \leftarrow lca_{tree}(ref, neg)$ &emsp;&emsp;&emsp;&emsp; $\triangleright \mathcal{O}(1)$ with [3]
7: &emsp; $reg \leftarrow reg + \max(0, \alpha + u(\sigma(pass\_node_1)) - u(\sigma(pass\_node_2)))$ &emsp; $\triangleright \mathcal{O}(1)$

8: **return** $reg$

---

---

**Algorithm 3** Dasgupta cost function (11)

---

**Require:** Graph $\mathcal{G} = (V, E)$ with ultrametric $u$
**Require:** Output Dapgusta cost function value
1: $soft\_cardinal \leftarrow$ soft-cardinal$((\mathcal{G}, u), tree)$ &emsp;&emsp; $\triangleright$ Algorithm 4 $\mathcal{O}(N^2)$
2: preprocess l.c.a on $tree$ &emsp;&emsp;&emsp;&emsp;&emsp;&emsp; $\triangleright \mathcal{O}(N)$ with [3]
3: $cost \leftarrow 0$ &emsp;&emsp;&emsp;&emsp;&emsp;&emsp;&emsp;&emsp;&emsp; $\triangleright \mathcal{O}(1)$
4: **for each** edge $e_{xy} \in E$ **do** &emsp;&emsp;&emsp;&emsp;&emsp;&emsp; $\triangleright \mathcal{O}(N)$
5: &emsp; $pass\_node \leftarrow lca_{tree}(x, y)$ &emsp;&emsp;&emsp;&emsp;&emsp; $\triangleright \mathcal{O}(1)$ with [3]
6: &emsp; $cost \leftarrow cost + soft\_cardinal(pass\_node)/u(e_{xy})$ &emsp;&emsp; $\triangleright \mathcal{O}(1)$

7: **return** $cost$

---

with Algorithm 1, the size of the nodes of the tree can be computed recursively from leaves to root in linear time. Note that, on line 9, the child of $y$ that does not contain $x$ can easily be determined by remembering the previous node of the "for each" loop: it is the sibling of the latter.

## 2.1  Framework validation

All the tests in the comparison with the CUCP algorithm (Section 4) were conducted on a computer with an Intel I7 4 cores processor and 16 GB of memory. For the optimization, we use the AMSGrad variation [4] of the ADAM method with step-size $0.01$. Our implementation of the proposed algorithms are all single threaded.

For each of the 52 test instances, the values $J_{\mathrm{closest}}$ obtained at each iteration of Algorithm 1 were normalized between 0 (lowest achieved cost for this instance) and 1 (highest cost reached for this instance). Figure 2 shows the mean-normalized convergence curve (with its standard deviation). We can see, that the convergence rate appears to be very good and smooth in practice. The convergence is usually reached within a bit more than a hundred iterations.

## 3  Illustrative examples

Figure 3 shows more illustrative examples of hierarchical clustering. For each dataset, we build a 5-nearest-neighbor graph, to which we add the edges of a minimum spanning tree to ensure the connectivity. Then, we perform hierarchical clustering on this graph, and we threshold the resulting ultrametric at the prescribed number of clusters. The column "Closest" is the solution to Algorithm 1 with the cost function $J_{\mathrm{closest}}$. The column "Closest+Size" is the solution to Algorithm 1 with the cost function $J_{\mathrm{closest}} + \lambda J_{\mathrm{size}}$ and $\lambda = 0.1$, where the regularization is only applied to the top-10 dendogram nodes. The column "Closest+Triplet" is the solution to Algorithm 1 with the cost function $J_{\mathrm{closest}} + \lambda J_{\mathrm{triplet}}$, $\lambda = 1$, and $\alpha = 3$. The column "Dasgupta" is the solution to Algorithm 1 with the cost function $J_{\mathrm{Dasgupta}}$. For the optimization, we use the AMSGrad variation [4] of the ADAM method with step-size $0.1$.

---
**Algorithm 4** Soft-cardinal function (16)

---
**Require:** Graph $\mathcal{G} = (V, E)$ with ultrametric $u$
**Require:** Single linkage clustering $tree$ on $(\mathcal{G}, u)$
**Require:** Output the soft-cardinal of non leaves node of $tree$
 1: $area \leftarrow$ cardinal of each node of $tree$             $\triangleright \mathcal{O}(N)$
 2: preprocess l.c.a on $tree$             $\triangleright \mathcal{O}(N)$ with [3]
 3: **for each** non leaf node $n$ of $tree$ **do**             $\triangleright \mathcal{O}(N)$
 4:     $pass\_edge \leftarrow \sigma(n)$             $\triangleright \mathcal{O}(1)$
 5:     $alt\_n \leftarrow u(pass\_edge)$             $\triangleright \mathcal{O}(1)$
 6:     $soft\_cardinal(n) \leftarrow 2 \times \text{sigmoid}(alt\_n)$             $\triangleright \mathcal{O}(1)$
 7:     **for each** vertex $x$ of $pass\_edge$ **do**             $\triangleright \mathcal{O}(1)$
 8:        **for each** ancestor $y$ of $x$ **do**             $\triangleright \mathcal{O}(N)$
 9:           $c\_other \leftarrow$ child of the $y$ that does not contain $x$             $\triangleright \mathcal{O}(1)$
10:           $contrib\_y \leftarrow area(c\_other) \times \text{sigmoid}(alt\_n - u(\sigma(()y)))$             $\triangleright \mathcal{O}(1)$
11:           $soft\_cardinal(n) \leftarrow soft\_cardinal(n) + contrib\_y$             $\triangleright \mathcal{O}(1)$
12: **return** $soft\_cardinal$

---

Figure 2: Mean-normalized convergence curve of the proposed approach and standard deviation.

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

Figure 3: Illustrative examples of hierarchical clustering. *Top rows*: Ultrametrics fitted to the input graph (only the top-30 non-leaf nodes are shown in the dendrograms). *Bottom rows*: Assignments obtained by thresholding the ultrametrics at two, three, or four clusters.

(a) Graph/Labels  (b) Average link  (c) Closest  (d) Closest+Size  (e) Closest+Triplet  (f) Dasgupta