[Reviews · NeurIPS 2019]

Reviewer 1



The paper introduces a new algorithm to learn hierarchical clusters based on fitting an ultrametric to the data. This is achieved by transforming an optimization problem over the constrained set of ultrametrics to one over an unconstrained set of weighted graphs, with the ultrametric constraint implicitly satisfied by the cost function. The idea of transforming the ultrametric fitting in a way conducive to gradient based methods is novel. It allows many ideas for regularization to modify hierarchical clusters in a way that standard hierarchical clustering approaches e.g. hierarchical agglomerative clustering does not allow as easily. The paper then demonstrates the flexibility of this new formulation by showing how different regularizations and data fidelity terms can be incorporated into the cost function. For example, noticing that in a vanilla MSE cost, small clusters appear high up in the dendrogram, the paper proposes penalizing such occurrences to produce more balanced hierarchies. Another example is the ability to allow semi-supervision in the form of specifying triplets. These cost functions are then showed to be approximated by differentiable functions and (in most cases) efficiently computed in a gradient descent framework. Finally, experimental results validate both the framework (using the CUCP algorithm as baseline) for runtime and accuracy and show that the quality of the clusters are comparable to standard algorithms (Ward and semi-supervised-SVM). The paper is well written and organized. It potentially opens new directions for exploring hierarchical clustering algorithms through the new ultrametric fitting approach.

Reviewer 2



The paper proposes an optimization framework for ultrametric fitting in the context of hierarchical clustering. The main idea of the paper is to use a min-max operator directly in a cost function defined for a given edge-weighted graph in order to find the optimal ultrametric for a dissimilarity graph. Two different ultrametric related cost functions and two different regularizations are used. Experimental evaluations on superpixel adjacency graphs from the datasets of the LIBSVM webpage illustrate the performance of this hierarchical clustering approach compared to state-of-the-art unsupervised Ward method and semi-supervised SVM. Strengths: - This paper address an interesting problem of hierarchical cluster analysis by using an optimizations framework for ultrametric fitting. - The usage of the min-max operator in order to solve the optimization problem of finding a best ultrametric has not been studied so far. - Paper is overall well written with a good structure. However, some details should be clarified (see below). - The authors make good use of already existing cost functions for evaluating an ultrametric for their purposes. - The experimental part gives a good overview of the comparison to the state-of-the-art agglomerative methods. Weaknesses: 1.) l Line 8,56,70,93,: Usage of the word equivalent. I would suggest a more cautious usage of this word. Especially, if the equivalence is not verified. 2.) A differentiation between the ultrametric d and the ultrametric u would make their different usages clearer. 3.) Line 186 ff. : A usage of the subdominant ultrametric for the cluster-size regularization, would make the algorithms part more consistent with the following considerations in this paper. 4) The paper is not sufficiently clear in some aspects (see below for a list of questions) Overall, I have the impressions the the weaknesses could be fixed until the final paper submission.

Reviewer 3



Originality: For the aforementioned contributions, I believe this work provides a creative, unique approach to this problem. Quality: I believe this paper to be technically sound, a complete work that presents interesting approaches for hierarchical clustering. Clarity: The paper is written well and clearly explains the approach. But there were a some details that I thought could have been made clearer in both the presentation and in the experiments. Unless I’ve missed something, I think that it would be good to more clearly state the process (and its complexity) of going from the ultrametric fit to data to a dendrogram. I believe this can be done using the LCA function applied to all pairs of points / the Jacobian of \phi. If there is no room in the body of the paper, I believe it could be included in the supplementary material. I also think the experiments could better show the effectiveness of the approach constructing high quality hierarchical clusterings. I believe a more clear experiment would compare their optimization of Dasgupta’s cost to alternative approaches to illustrate which approaches find clusterings with lower costs. Significance: I believe the work introduces novel methodology and a creative approach in a relatively unexplored space of gradient-based methods for hierarchical clustering. The methodology is clearly expressed. However, I do feel that the experiments of the paper are a bit weak and could be made stronger as mentioned in the clarity section. The work compares both hierarchical clustering objectives and optimization methods, but it would be stronger if the experiments compared the proposed method to alternative approaches that also optimize the Dasgupta cost function. I also think the paper would benefit from a lengthier discussion of the merits of gradient-based approaches for ultrametric fitting (and perhaps further experiments to support this). The paper does not give theoretical results.

[Author Response · NeurIPS 2019]

We thank the reviewers for the time they spent evaluating our manuscript and for their valuable comments. All the
answers given in this document will be incorporated in the manuscript and/or in its appendix. In the following we took
the liberty to group and reword some of the reviewers comment (in blue italic) to save space.

**General answer on the usefulness of gradient descent, its theoretical guarantees, and its scalability.**

The proposed gradient-based approach has two important advantages: (i) it is extremely flexible, allowing us to easily
incorporate new cost terms; (ii) it provides a generic "ultrametric layer" that can be used to build end-to-end learning
pipelines, for example with methods for graph embedding in hyperbolic spaces (Nickel and Kiela, NeurIPS 2017).
We agree that having theoretical guarantees would be a big plus. However, we know, in particular with recent advances
on deep neural networks, that obtaining any insight toward theoretical guarantees with gradient descent algorithms
applied to nonconvex problems is extremely challenging, and we hope to be able to get theoretical results in future
works. In this regard, apart from what mentioned in the manuscript, an interesting line of research arises from recent
works on the implicit bias of gradient descent by Soudry *et al.* (ICLR 2018) and Ji *et al.* (COLT 2019).
As for scalability, the bottleneck of our method is the single-linkage algorithm. One way to address this issue (up to
a certain limit) would be to use recent progresses on parallel single-linkage algorithms (out-of-core algorithms can
process billions of vertices). Nonetheless, true scalability will require to adapt mini-batch gradient descent to ultrametric
fitting, which is a topic that we are currently working on. Similarly to Monath *et al.* (NeurIPS 2017), our idea consists
of only updating sub-trees at each iteration. However, this leads to a biased approximation of the gradient, that we plan
to correct by maintaining an exponentially-weighted average of the Jacobian matrix, like in Pfau *et al.* (ICLR 2019).
Given the significant body of additional material, we feel that this topic is best left to a future publication.

*R1: Experiments on larger datasets for comparison of hierarchical clustering quality.* As scalable solutions for dealing
with massive datasets require nontrivial extensions, we believe that they deserve a separate publication.

*R2: Line 8,56,70,93: I would suggest a more cautious usage of the word "equivalent".* Your are right that the transformed
problem is not strictly equivalent to the original one, we will thus remove the world "equivalent". Moreover, a proof of
equivalence between ultrametric and hierarchical clustering can be found in [29, Th. 9].

*R2: A differentiation between the ultrametric d and the ultrametric u would make their different usages clearer.* We
will improve the beginning of Section 2 and its illustration in Figure 1 in order to better explain the difference and the
importance of modeling the ultrametric $d_u$ by its compact "restriction" $u$ defined only on the edges of the graph.

*R2: Line 186 ff.: A usage of the subdominant ultrametric for the cluster-size regularization, would make the algorithms
part more consistent with the following considerations in this paper, and One more sentence to the relationship between
(13) and (17) would make the transition with the use of the subdominant ultrametric compared to before clearer.* We
will emphasize how Eq (14) can be used to rewrite cluster size regularization and Dasgupta's cost to suit the proposed
optimization framework. In particular, we will give the full derivation of Eq (13) to Eq (17) in the appendix.

*R2: How is the weighting $\lambda$ of the regularization term chosen? Were also other values used?* We tried several values on
an exponential scale and we observed that the constant value reported in the article was sufficient for all the experiments.

*R2: Line 149: $J_{Dasgupta}$ calculated with the Heaviside function H or the sigmoid function? and Equation (17): what
is $\ell$?* A continuous approximation of the Heaviside function is mandatory for Dasgupta's cost, as the cluster sizes is the
only element of the cost that depends of the ultrametric $u$. With the Heaviside function, the derivative of Dasgupta's
cost will be null almost everywhere. The undefined symbol $\ell$ in (17) was indeed meant to refer to the sigmoid function.

*R2: Notations and clarifications. 1) Line 128: labels $C_v$, why $C$ is used? $C$ already used for cycles in $G$. 2) Algorithm
2 Title: defined in (14) instead of (5)?, 3) Line 144: $x \in n$. $x$ is a node and $n$ also? 4) Line 149: Was the Dasgupta cost
function for the visualization in Figure 2 also combined with a regularization as in the experimental part?* 1) We will
replace $C_v$ with $\mathcal{L}_v$ (label of $v$) to avoid any confusion. 2) Yes, the algorithm computes the min-max operator defined
in (5) with the formulation given in (14). 3) You are right, nodes are subsets of $V$, so $x \in V$ is an element of a node $n$,
and the singleton $\{x\}$ is a leaf node. 4) No, Dasgupta's cost was used without regularization in all the toy examples.

*R3: I think that it would be good to more clearly state the process (and its complexity) of going from the ultrametric
fit to data to a dendogram. and Clarifications of how a hierarchical clustering is extracted from a fit ultrametric.*
The dendrogram associated to a given ultrametric is a byproduct of Algorithm 2 (tree computed on line 2). The final
dendrogram is thus obtained for free at the end of Algorithm 1, when computing the min-max operator on the estimated
edge weights. Thresholding the dendrogram node altitudes yields a flat clustering. In practice, one can efficiently find
the threshold that would produce a given number of clusters by browsing the dendrogram from the root to the leaves.

*R3: Experiments that compare the optimization of hierarchical clustering methods wrt Dasgupta's cost.* The figure beside compares Dasgupta's cost obtained with Linkage++ [11] and the proposed relaxed cost function $J_{Dasgupta}$ on 50 random graphs with cosine similarities (blobs-like structure with 1024 points organized in 16 clusters). These preliminary results shows that the proposed relaxation is likely close to the exact Dasgupta's cost. Note that this is consistent with Fig 1(a) in [11]. We plan to further explore this question with real data in the final version.



[Meta-Review · NeurIPS 2019]

The authors develop a nice method for fitting a hierarchical clustering by solving an optimization problem that leads to an ultrametric (and thus a hierarchical clustering). The reviewers are in agreement that this work should be accepted.